# Sesquiterpenoids from the Mangrove-Derived *Aspergillus ustus* 094102

**DOI:** 10.3390/md20070408

**Published:** 2022-06-22

**Authors:** Pengyan Gui, Jie Fan, Tonghan Zhu, Peng Fu, Kui Hong, Weiming Zhu

**Affiliations:** 1Key Laboratory of Marine Drugs, Ministry of Education of China, School of Medicine and Pharmacy, Ocean University of China, Qingdao 266003, China; 11180811003@stu.ouc.edu.cn (P.G.); fj900622@163.com (J.F.); sdueduzth@126.com (T.Z.); fupeng@ouc.edu.cn (P.F.); 2Qingdao Marine Technical College, Qingdao 266590, China; 3College of Civil Engineering and Architecture, Shandong University of Science and Technology, Qingdao 266590, China; 4Laboratory for Marine Drugs and Bioproducts of Qingdao National Laboratory for Marine Science and Technology, Qingdao 266237, China; 5Key Laboratory of Combinatorial Biosynthesis and Drug Discovery, Ministry of Education and School of Pharmaceutical Sciences, Wuhan University, Wuhan 430071, China

**Keywords:** drimane sesquiterpenoids, absolute configuration, antiproliferation, *Aspergillus ustus*, mangrove-derived fungus

## Abstract

Four new drimane sesquiterpenoids (**1**–**4**) and three known ones (**5**–**7**) were isolated from the fermentation broth of the mangrove-derived *Aspergillus ustus* 094102. Compound **5** was further resolved as four purified compounds **5a**–**5d**. By means of extensive spectroscopic and ECD analysis as well as the chemical transformation, their structures were identified as (2*R*,3*R*,5*S*,9*R*,10*S*)-2,3,9,11-tetrahydroxydrim-7-en-6-one (ustusol F, **1**), (2*R*,3*R*,5*R*,9*S*,10*R*)-2,3,11-trihydroxydrim-7-en-6-one (9-deoxyustusol F, **2**), (3*S*,5*R*,9*R*,10*R*)-3,11,12-trihydroxydrim-7-en-6-one (ustusol G, **3**), (5*S*,6*R*,9*S*,10*S*, 11*R*,2′*E*,4′*E*)-(11-dideoxy-11-hydroxystrobilactone A-6-yl)-5-carboxypenta-2,4-dienoate (ustusolate H, **4**), ((5*S*,6*R*,9*S*,10*S*)-strobilactone A-6-yl) (2*E*,4*E*)-6,7-dihydroxyocta-2,4-dienoate (ustusolate I, **5**), (2′*E*,4′*E*;6′,7′-*erythro*)-ustusolate I (**5a**) and (2′*E*,4′*E*;*ent*-6′,7′-*erythro*)-ustusolate I (**5b**), (2′*E*,4′*E*,6′*R*,7′*R*)-ustusolate I (**5c**) and (2′*E*,4′*E*,6′*S*,7′*S*)-ustusolate I (**5d**), (5*S*,6*R*,9*S*,10*S*,2′*E*,4′*E*)-(strobilactone A-6-yl)-5-carboxypenta-2,4-dienoate (ustusolate J, **6**), and (2*S*,5*S*,9*R*,10*S*)-2,9,11-trihydroxydrim-7-en-6-one (ustusol B, **7**), respectively. Compound **5** showed antiproliferation against the human tumor cells CAL-62 and MG-63 with the IC_50_ values of 16.3 and 10.1 µM, respectively.

## 1. Introduction

As well as we know, microbial metabolites are an important source of drug discovery and development [1]. However, with the deepening of research, many strains in the conventional environment have been repeatedly studied, resulting in the increase of the recurrence rate of known compounds and the decrease of the occurrence rate of new bioactive compounds [2]. Mangrove fungi have attracted much attention because of their special growth environment and unique metabolic mechanism, resulting in the diversity of the structure and bioactivity of their secondary metabolites, which has become a new hotspot in drug development [3,4]. Our previous work reported 9 drimane sesquiterpenoids, 8 benzofurans [4], and 18 ophiobolins [5] from mangrove-derived fungus *Aspergillus ustus* 094102, among which ustusorane E and ustusolate E exhibited cytotoxic activity against the HL-60 cells with IC_50_ values of 0.13 and 9.0 μM, respectively [4]. In addition, more than 50 drimane sesquiterpenoids have been reported from fungi, including cytotoxic strobilactones A and B from *Strobilurus ohshimae* [6], (6-strobilactone-B) ester of (*E*,*E*)-6-oxo-2,4-hexadienoic acid from marine sponge-derived *A*. *ustus* [7], (6-strobilactone-B) ester of (*E*,*E*)-6-carbon-7-hydroxy-2,4-octadienoic acid from mangrove-derived *A*. *ustus* [8], synergistic antibacterial ustusoic acid B from *A*. *ustus* [9], and ET-1 binding inhibitory (2′*E*,4′*E*,6′*E*)-6-(l’-carboxyocta-2′,4′,6′-triene)-9-hydroxydrim-7-ene-l l-al from *A*. *ustus* var. *pseudodeflectus* [10], etc. In order to further explore the new drimane sesquiterpenoids produced by *A*. *ustus* strain 094102, we continued to study its secondary metabolites. As a result, we isolated and identified four new drimane sesquiterpenoids (**1**–**4**), as well as three known analogues, (strobilactone A-6-yl) (2*E*,4*E*)-6,7-dihydroxyocta-2,4-dienoate (**5**) [7] that were further isolated as four isomers **5a**–**5d**, mono(6-strobilactone A) ester of (*E*,*E*)-2,4-hexadienedioic acid (**6**) [7], and 2α,9α,11-trihydroxy-6-oxodrim-7-ene (**7**) [7]. The structures elucidation including absolute configurations and the antiproliferative activity will be discussed here.

## 2. Results and Discussion

The bioactive EtOAc extract of the fermentation broth of the mangrove-derived fungus *Aspergillus ustus* 094102 was chromatographed on silica gel, Sephadex LH-20, and preparative HPLC columns to give compounds **1**–**7** (Figure 1).

Compound **1** was obtained as a colorless oily solid. Its molecular formula was determined as C_15_H_24_O_5_ based on the high-resolution mass spectrometry (HRMS, ESI-Orbitrap) peak at *m*/*z* 285.1694 [M+H]^+^ or 283.1547 [M–H]^−^ (Appendix A), indicating 4 index of hydrogen deficiency (IHD). The IR spectrum at ν_max_ 3399 and 1663 cm^−1^ (Appendix A), corresponded to a hydroxy and an *α*,*β*-unsaturated carbonyl group, respectively. The ^1^H-NMR data (Table 1, Appendix A) of **1** revealed four tertiary methyl groups at *δ*_H_ 1.04 (s, H-13/15), 1.14 (s, H-14) and 1.96 (s, H-12), an oxymethylene signal at *δ*_H_ 3.53/3.64 (d/d, H-11), a methylene signal at *δ*_H_ 1.85/1.69 (dd/t, H-1), one olefinic proton signal at *δ*_H_ 5.61 (d, H-7), three methine signals at *δ*_H_ 3.47 (dt, H-2), 2.67 (d, H-3) and 2.81 (s, H-5), as well as four exchangeable proton signals at *δ*_H_ 4.48 (HO-3/2), 4.91 (HO-11) and 5.06 (HO-9). The ^13^C-NMR and DEPT data (Table 1, Appendix A) of **1** revealed 15 carbon signals, including a ketone carbonyl signal at *δ*_C_ 199.2 (C-6), two olefinic carbons at *δ*_C_ 128.2/157.4 (C-7/C-8), four methyl signals at *δ*_C_ 16.7/19.1/19.2/29.3 (C-15/C-13/C-12/C-14), two methylenes at *δ*_C_ 38.6/61.9 (C-1/C-11), three methines at *δ*_C_ 55.0/66.4/81.6 (C-5/C-2/C-3) and three nonhydrogenated carbons at *δ*_C_ 37.8/45.4/74.6 (C-4/C-10/C-9). These NMR data (Table 1) were closely related to those of 3*β*,9*α*,11-trihydroxydrim-7-en-6-one (that is 3*β*,9*α*,11-trihydroxy-6-oxodrim-7-ene [7]), indicating the presence of a drimane sesquiterpene skeleton. The key difference was that compound **1** possessed an additional hydroxy group that resided at C-2 of ring A. On the basis of correlations in the COSY experiments between HO-3/H-3, H-3/H-2/H-1 and HO-11/H-11, as well as the key HMBC correlations from H-1 to C-5/C-10/C-13, H-3 to C-4/C-14/C-15, H-5 to C-4/C-6/C-9/C-10/C-13/C-14/C-15, H-7 to C-5/C-9/C-12, H-11 to C-8/C-9/C-10, H-12 to C-7/C-8/C-9, H-13 to C-5/C-9/C-10, H-14 to C-3/C-4/C-5/C-15, and H-15 to C-3/C-4/C-14 (Figure 2 and Appendix A) further confirmed the constitution of **1** (Figure 1). The relative configuration was deduced from the NOESY spectrum (Figure 3 and Appendix A), which showed correlations of H-1*α* to H-3/H-5/HO-9, H-11 to H-1*β*/H-2/H-13, and H-2 to H-13 indicated *cis*-orientation of HO-2/H-5/HO-9, and H-2/HO-3/CH_3_-13/CH_2_-11, and a *trans*-fused decalin nucleus. The absolute configuration of **1** was determined by its ECD spectrum. On the basis of the octant rule for cyclohexenones [11,12,13], the positive Cotton effect at λ_max_ 336 nm (∆ε + 8.4) and the negative Cotton effect at λ_max_ 240 nm (∆ε − 41.3) (Figure 3 and Appendix A) indicated the (2*R*,3*R*,5*S*,9*R*,10*S*)-configuration, consistent with the core configuration of the drimane sesquiterpene, 9*α*,11-dihydroxydrim-7-en-6-one (that is 6-oxo-7-drimen-9*α*,11-diol [14]), whose absolute configurations have been established by chemical synthesis. Therefore, compound **1,** which we named ustusol F, was determined as (2*R*,3*R*,5*S*,9*R*,10*S*)-2,3,9,11-tetrahydroxydrim-7-en-6-one.

Compound **2** was obtained as a light-yellow oil. Its molecular formula was determined as C_15_H_24_O_4_ based on the HRESIMS peak at *m*/*z* 269.1751 [M+H]^+^ (Appendix A). The similar IR and UV absorptions to those of **1** implied that they shared the same molecular skeleton (Appendix A). The 1D NMR data (Table 1, Appendix A) were also similar to **1** except for a methine signal at *δ*_C/H_ 57.1/2.29 which replaced the nonhydrogenated oxycarbon at *δ*_C_ 74.6, the disappearance of a hydroxy signal at *δ*_H_ 5.06 (HO-9), and the changes of chemical shifts around C-9. These data combined with the 16 amu less of molecular weight than **1** revealed compound **2** as the 9-deoxy derivative of compound **1**. Key COSY of H-9/H-11/HO-11 and HMBC of H-11 to C-8 and C-10 and H-9 to C-10 (Figure 2, Appendix A) supported the inference. The same relative configuration to **1** was deduced from the NOESY correlations of H-2 (*δ*_H_ 3.45) to H-13 (*δ*_H_ 0.88), H-15 (*δ*_H_ 1.03) and H-1*β* (*δ*_H_ 2.09), H-1*α* (*δ*_H_ 1.33) to H-3 (*δ*_H_ 2.75), H-5 (*δ*_H_ 2.22) and H-9 (*δ*_H_ 2.29), and H-3 to H-14 (*δ*_H_ 1.12) (Figure 3 and Appendix A). The absolute configuration of the *threo*-2,3-diol in **2** was assigned by a dimolybdenum-induced ECD method [15,16]. Upon addition of Mo_2_(OAc)_4_ to a DMSO solution of compound **2**, a chiral dimolybdenum complex was generated in situ as an auxiliary chromophore. Because the contribution from the inherent ECD was subtracted to give the induced ECD of the complex, the observed sign of the Cotton effect in the induced spectrum originates solely from the chirality of the *ortho*-diol moiety expressed by the sign of the O–C–C–O torsion angle. The positive Cotton effect at λ_max_ 332 (∆*ε* + 6.8) nm (Appendix A) permitted us to assign the (2*R*,3*R*)-configuration on the basis of Snatzke’s empirical rule [15]. In addition, compounds **1** and **2** also showed a similar ECD Cotton effect, indicating the same absolute configuration. Thus, compound **2,** which we named 9-deoxyustusol F, was determined as (2*R*,3*R*,5*R*,9*S*,10*R*)-2,3,11-trihydroxydrim-7-en-6-one.

Compound **3** was obtained as a colorless oily solid. Its molecular formula was determined as C_15_H_24_O_4_ based on the HRESIMS peak at *m*/*z* 269.1750 [M+H]^+^ (Appendix A), indicating an isomer of **2**. Similar 1D NMR data (Table 1, Appendix A) with **2** were observed. In addition, a methylene signal (*δ*_H/C_ 1.47/26.7) and an oxymethylene signal (*δ*_H/C_ 4.19/4.26/61.3) replaced the methyl signal (*δ*_H/C_ 1.98/21.5) and oxymethine signal (*δ*_H/C_ 3.45/66.1). Key COSY of H-1/H_2_-2/H-3 as well as the HMBC of H_2_-12 (*δ*_H_ 4.19/4.26) to C-8 (*δ*_C_ 162.3), H-7 (*δ*_H_ 5.96) to C-12 (*δ*_C_ 61.3) and H_2_-2 (*δ*_H_ 1.47) to C-4 (*δ*_C_ 37.5) and C-10 (*δ*_C_ 41.7) revealed that 2-OH was moved to C-12 to form 2-CH_2_ and 12-CH_2_OH, respectively (Figure 2, Appendix A). The relative configuration of compound **3** was deduced from the NOE difference (NOEdiff) experiment. NOEdiff of **3** showed that H-5 (*δ*_H_ 2.15) and H-1a (*δ*_H_ 1.44) were enhanced after the irradiation of H-9 (*δ*_H_ 2.31), while H-3 (*δ*_H_ 3.02) and H-9 (*δ*_H_ 2.31) were enhanced after the irradiation of H-5. The NOE enhancements of H-3 (*δ*_H_ 3.02) and H-5 (*δ*_H_ 2.15) were also observed after H-14 (*δ*_H_ 1.10) was irradiated, while H-13 (*δ*_H_ 0.80) was enhanced after the irradiation of H-15 (*δ*_H_ 0.99). H-1b (*δ*_H_ 1.88) and H-15 was enhanced after the irradiation of H-13 (Appendix A). These NOE data indicated the *cis*-orientation of H-3, H-5, H-9 and H-14 as well as H-13 and H-15, indicating the same relative configuration of **3** to **2** in the chiral centers of C-3, C-5, C-9, and C-10. The similar ECD spectrum to that of **2** implied the same absolute configuration, which was confirmed by octant rule for cyclohexanone [11,12,13], the positive Cotton effect at λ_max_ 335 nm (∆ε + 10.6) and the negative Cotton effect at λ_max_ 241 nm (∆ε – 19.1) (Figure 4 and Appendix A). Accordingly, compound **3**, which we named ustusol G, was elucidated as (3*S*,5*R*,9*R*,10*R*)-3,11,12-trihydroxydrim-7-en-6-one.

Compound **4** was obtained as a colorless solid. Its molecular formula was determined as C_21_H_28_O_7_ based on the HRESIMS peak at *m*/*z* 391.1762 [M–H]^–^, indicating 8 HIDs (Appendix A). The IR spectrum showed absorption bands of hydroxyl and conjugated carbonyl at ν_max_ 3434 and 1696 cm^−1^ (Appendix A), respectively. The 1D NMR spectra of **4** (Table 2, Appendix A) were very similar to those of (2*E*,4*E*)-(strobilactone A-6-yl)-5-carboxypenta-2,4-dienoate (that is mono(6-strobilactone B) ester of (*E*,*E*)-2,4-hexadienedioic acid [7]), which we named ustusolate J (**6**) for convenience, suggesting that they shared the same molecular scaffold. The only difference was a replacement of the lactone carbonyl signal (*δ*_C_ 174.6 in **6**) by the hemiacetal methine group (*δ*_C/H_ 97.4/5.20 in **4**). In addition, the chemical shifts for C-9 and C-7 have a great increase and decrease, respectively (Table 2 and Appendix A). These data combined with a 2 amu more than **6** suggested that the γ-lactone of **6** was reduced to the corresponding hemiacetal in **4**. The key HMBC correlations from hemiacetal proton (*δ*_H-11_ 5.20) to C-9 (*δ*_C_ 76.4)/C-10 (*δ*_C_ 38.0)/C-12 (*δ*_C_ 65.8), from H-12 (*δ*_H_ 4.08/4.38) to C-7 (*δ*_C_ 117.0)/C-8 (*δ*_C_ 143.2)/C-9/C-11 (*δ*_C_ 97.3), and from H-7 (*δ*_H_ 5.49) to C-5 (*δ*_C_ 45.1)/C-9 verified the deduction (Figure 2 and Appendix A). Compound **4** displayed the key NOESY correlations of H-6 (*δ*_H_ 5.58) with H-5 (*δ*_H_ 2.07) and H-14 (*δ*_H_ 0.91), H-5 with H-1b (*δ*_H_ 1.86) and H-2a (*δ*_H_ 1.42), H-11 (*δ*_H_ 5.20) with H-1a (*δ*_H_ 1.22), as well as H-13 (*δ*_H_ 1.12) with H-2b (*δ*_H_ 1.58) (Figure 3 and Appendix A), indicating *cis*-orientation of H-5 with H-6 and *trans*-orientation of H-5 with H-11 and H-13 which is the same relative configuration of **4** to **1** and **6** in the decalin (decahydronaphthalene) nucleus. The same relative configuration of HO-9 was deduced from the same biosynthetic pathway to those of compounds **1** and **5**–**7**. Subsequently, the same ECD pattern of **4**–**6** (Appendix A) implied the same absolute configuration of the drimane nucleus. Compound **4,** which we named ustusolate H, was thus elucidated as (5*S*,6*R*,9*S*,10*S*,11*R*,2′*E*,4′*E*)-(11-deoxy-11-hydroxystrobilactone A-6-yl)-5-carboxypenta-2,4-dienoate.

Compound **5** was obtained as a yellow oil. Its molecular formula was determined as C_21_H_28_O_7_ based on the ESIMS peak at *m*/*z* 419.1 for [M–H]^–^ and *m*/*z* 464.9 for [M + HCO_2_]^–^(Appendix A), indicating 8 HIDs. A literature search verified that the constitution (planar structure) of compound **5** was the same as the (strobilactone A-6-yl) (2*E*,4*E*)-6,7-dihydroxyocta-2,4-dienoate (that is (6-strobilactone-B) esters of (*E*,*E*)-6,7-dihydroxy-2,4-octadienoic acid [7]), for almost the same NMR data. However, four sets of ^13^C NMR signals of compound **5** (Appendix A) for the side chain at *δ*_C_ 165.51/165.50/165.49/165.47 (C-1′), 120.03/120.99/119.95/119.90 (C-2′), 145.41/145.37/145.34/145.26 (C-3′), 127.54/127.35/127.16/126.98 (C-4′), 146.18/146.12/145.48/145.45 (C-5′), 75.16/75.00/74.64/74.46 (C-6′), 69.64/69.62/69.33/69.32 (C-7′), and 19.34/19.26/18.26/18.24 (C-8′) were observed, indicating four stereoisomers of **5** resulted from the *ortho*-diol chiral centers of the side chain. With the help of HPLC, compound **5**, which we named ustusolate I for convenience, was confirmed to have four baseline-separated peaks, then purified **5a**, **5b**, **5c**, and **5d** were obtained (Appendix A). The NMR differences of **5a**–**5d** were concentrated in the side chains (Table 3 and Table 4, Appendix A), and indicated that compounds **5a** and **5b**, **5c**, and **5d** were two pairs of enantiomers of the *ortho*-diol in the side chain. To elucidate the relative configuration of 6′,7′-diol moiety, the acetonide (**5e**) was prepared from **5a** (Figure 4). The 1D and 2D NMR spectra (Table 3 and Table 4, Appendix A), as well as the NOESY correlations of H-5′ (*δ*_H_ 6.24)/H_3_-8′ (*δ*_H_ 1.01) and H_3_-11′ (*δ*_H_ 1.40), H-6′ (*δ*_H_ 4.62)/H-7′ (*δ*_H_ 4.34) and H_3_-10′ (*δ*_H_ 1.28) in **5e** (Figure 3 and Appendix A) clearly suggested an *erythro*-6′,7′-diol in **5a** and **5b**, and a *threo*-6′,7′-diol in **5c** and **5d** was accordingly elucidated. This conclusion is consistent with the chemical shift rule of methyl carbon (*δ*_CH3_) for 1-methyl-1,2-diol by chemical synthesis, that is 18.1–18.6 and 19.1–19.6 ppm for *threo*- and *erythro*-1,2-diol, respectively [17]. The absolute configuration of the *threo*-6′,7′-diol in **5c** and **5d** was assigned by a dimolybdenum-induced ECD method [15,16] in the same manner as that of compound **2**. Upon addition of Mo_2_(OAc)_4_ to a solution of compounds **5****c** and **5d** in DMSO, a chiral dimolybdenum complex was generated in situ as an auxiliary chromophore. According to the negative ECD Cotton effects of **5c** at λ_max_ 303 (Δ*ε* – 7.9) nm and the positive ECD Cotton effects of compound **5d** at λ_max_ 305 (Δ*ε* +2.37) nm (Appendix A), the absolute configuration of *threo*-6′,7′-diol in **5c** and **5d** were determined to be (6′*R*,7′*R*) and (6′*S*,7′*S*), respectively. Thus, the structure of **5c** and **5d** was unambiguously determined as (2′*E*,4′*E*,6′*R*,7′*R*)-ustusolate I (**5c**) and (2′*E*,4′*E*,6′*S*,7′*S*)-ustusolate I (**5d**), respectively. Unfortunately, the absolute configuration of compounds **5a** and **5b** were not determined yet in this paper, which we tentatively named (2′*E*,4′*E*;6′,7′-*e**rythro*)-ustusolate I (**5a**) and (2′*E*,4′*E*; *ent*-6′,7′-*e**rythro*)-ustusolate I (**5b**), respectively.

Compounds **6** and **7** which could be a 3-deoxy derivative of ustusol F (**1**) were identified by respective comparison of NMR data with those of mono(6-strobilactone-B) ester of (*E*,*E*)-2,4-hexadienedioic acid [7] and 2*α*,9*α*,11-trihydroxy-6-oxodrim-7-ene [7]. The same ECD pattern of compound **6** with **5** (Appendix A) and compound **7** with **1** (Appendix A) indicated they shared the same absolute configuration. Thus, compounds **6** and **7** were respectively identified as (5*S*,6*R*,9*S*,10*S*,2′*E*,4′*E*)-(strobilactone A-6-yl)-5-carboxypenta-2,4-dienoate (ustusolate J, **6**) and (2*S*,5*S*,9*R*,10*S*)-2,9,11-trihydroxydrim-7-en-6-one (ustusol B, **7**) in this paper. In addition, compound 7 showed almost the same NMR data as our previously reported ustusol B [4] (Appendix A) and displayed the same retention times in the co-HPLC (Appendix A). Thus, the structure of ustusol B was revised as structure **7,** which was named ustusol B.

The drimane sesquiterpenoids **1**–**7** were postulated to be biosynthesized from farnesyl-PP (**I**) which generated intermediate **II**, **III** and **IV** after cyclization and oxidation. The intermediates **II** and **III** were subjected to further oxidation to form compounds **1**, **2**, **3**, and **7**. The intermediate **II** was further oxidized to intermediate **IV**, and the latter was subjected to oxidation, hemi acetalization, and esterification to form compounds **4**, **5**, and **6** (Figure 5).

The antiproliferations of compounds **1**–**7** were evaluated against 29 human cancer cell lines and a normal cell line (the names of cell lines are listed in the Appendix A) by the cell counting Kit-8 (CCK-8) methods [18,19]. Only compound **5**, the mixture of **5a**/**5b**/**5c**/**5d**, showed antiproliferative activity against the human thyroid cancer cells (CAL-62) and human osteosarcoma cells (MG-63) with the IC_50_ values of 16.28 ± 1.01 and 10.08 ± 0.04 µM, respectively, while the pure compounds **5a**–**5d** were inactive (IC_50_ ≥ 50 µM). The IC_50_ values of doxorubicin (positive control) against CAL-62 and MG-63 were 0.062 ± 0.022 and 0.096 ± 0.012 µM, respectively. The bacteriostatic activities of compounds **1**–**7** against 6 human pathogenic bacteria and 6 aquatic pathogenic bacteria (the names are listed in the Appendix A) were tested by the diffusion method of filter paper, but no inhibition zone was observed at the concentration of 100 μg/mL for compounds **1**–**7**.

## 3. Experimental Section

### 3.1. General Experimental Procedures

Optical rotations were measured with a JASCO P-1020 digital polarimeter. UV data were recorded with a Beckman DU 640 spectrophotometer, and ECD data were collected using a JASCO J-715 spectropolarimeter. IR spectra were taken on a Nicolet NEXUS 470 spectrophotometer as KBr disks. ^1^H, ^13^C, DEPT, HMQC, HMBC, COSY, and NOESY NMR spectra were recorded on a JEOL JNM-ECP 600 spectrometer or a Bruker Avance 500 spectrometer in DMSO-*d*_6_ solution and were referenced to the corresponding residual solvent signals (*δ*_H/C_ 2.50/39.52 for DMSO-*d*_6_). HRESIMS spectra were collected using a Q-TOF Ultima Global GAA076 LC mass spectrometer. ESIMS data were measured using a Waters ACQUITY SQD 2 UPLC/MS system with a reversed-phase C18 column (ACQUITY UPLC BEH C18, 2.1 × 50 mm, 1.7 μm) at a flow rate of 0.4 mL/min. Semipreparative HPLC was performed using an ODS column (YMC- pack ODS-A, 10 × 250 mm, 5 μm, 4 mL/min) and a phenyl column (YMC-pack Ph, 10 × 250 mm, 5 μm, 4 mL/min). Vacuum–liquid chromatography (VLC) utilized silica gel H (Qingdao Marine Chemical Factory, Qingdao, China). TLC were carried out by plates precoated with silica gel GF254 (10–40 μm, Qingdao Marine Chemical Factory) and Sephadex LH-20 (Amersham Pharmacia Biotech, Buckinghamshire, UK) were used for column chromatography (CC).

### 3.2. Fungal Material

The mangrove fungal strain *A. ustus* 094102 was isolated from the rhizosphere soil of the mangrove plant *Bruguiera gymnorrhiza* grown in Wenchang, Hainan Province of China. It was identified according to the morphological characteristics and the ITS sequences [4,5].

### 3.3. Cultivation and Extraction

The fungus *A. ustus* 094102 was statically cultured at 25 °C for 28 days in one hundred 1000 mL conical flasks, each containing 300 mL of the liquid medium that was prepared by dissolving maltose (20 g), mannitol (20 g), glucose (10 g), monosodium glutamate (10 g), yeast extract (3 g), corn steep liquor (1 g), CaCO_3_ (2 g), KH_2_PO_4_ (0.5 g), MgSO_4_·7H_2_O (0.3 g), and sea salt (33 g) in 1 L of tap water (pH 7.0). The whole fermentation broth (30 L) was filtered by cheesecloth to separate the mycelia from the filtrate. The mycelia were extracted three times with an 80% volume of aqueous acetone. The acetone solution was concentrated under reduced pressure to give an aqueous solution. The aqueous solution was extracted three times with an equivalent volume of ethyl acetate (EtOAc), while the filtrate was extracted three times with an equivalent volume of EtOAc. All EtOAc extracts were combined and concentrated under vacuum to give 240 g of crude gum.

### 3.4. Purification

The crude gum (240 g) was separated into ten fractions (Fr1–Fr10) on a silica gel VLC column using a stepwise gradient elution of petroleum ether (PE), PE-CH_2_Cl_2_ (1:1–0:1) followed by CH_2_Cl_2_-MeOH (1:0–1:1). Fr9 (26 g) was fractionated on Sephadex LH-20, eluted with CH_2_Cl_2_-MeOH (1:1), to obtain three subfractions (Fr9.1–Fr9.3). Fr9.2 (8 g) was further separated into five subfractions (Fr9.2.1–Fr9.2.5) by VLC on the RP-18 column using a stepwise gradient elution of MeOH-H_2_O (9:1–1:1), among which the elution of 40% MeOH–H_2_O gave compound **7** (9.2 mg). Compounds **1** (6.2 mg, t_R_ 11.8 min) and **2** (32 mg, t_R_ 18.7 min) were obtained from Fr9.2.2 (1.7 g) by semipreparative HPLC over an ODS column eluting with 15% MeCN-H_2_O containing 0.5‰ Et_3_N. Fr7 (12 g) was fractionated on Sephadex LH-20, eluted with MeOH-CH_2_Cl_2_ (1:1), to obtain four subfractions (Fr7.1–Fr7.4). Fr7.4 (3.3 g) was further purified by semipreparative HPLC over an ODS column eluting with 40% MeCN-H_2_O containing 0.5‰ TFA (trifluoroacetic acid) to yield compound **4** (7.6 mg, t_R_ 16.5 min). Fr7.3 (1.3 g) was fractionated into four subfractions (Fr7.3.1–Fr7.3.5) on a RP-18 column using a stepwise gradient elution of MeOH-H_2_O (1:9–2:3). Fr7.3.2 (300 mg) was further separated by semipreparative HPLC on an ODS column eluted with 20% MeCN-H_2_O to yield compound **3** (3.1 mg, t_R_ 7.8 min). Fr6 (17.6 g) was further fractionated on Sephadex LH-20 eluted with MeOH-CH_2_Cl_2_ (1:1) to afford four subfractions (Fr6.1–Fr6.4). Fr6.2 (1.1 g) was further separated by semipreparative HPLC on an ODS column eluted with 40% MeCN-H_2_O containing 0.5‰ TFA to yield compound **6** (16.3 mg, t_R_ 15 min), while compound **5** (860 mg, t_R_ 17.0 min) was purified from Fr6.4 (9 g) by semipreparative HPLC on an ODS column eluted with 65% MeCN-H_2_O. Pure compounds **5a** (8.8 mg, t_R_ 39 min), **5b** (5.4 mg, t_R_ 42 min), **5c** (7.3 mg, t_R_ 44 min) and **5d** (6.8 mg, t_R_ 46 min) were obtained from compound **5** by a careful separation on an ODS column eluted with 50% MeOH-H_2_O.

### 3.5. The Preparation of Acetonide (***5e***) for Relative Configuration

According to our procedure [16], compound **5a** (5 mg) in acetone (3 mL) was added to the mixture of 2,2-dimethoxypropane (1 mL), pyridinium *p*-toluenesulfonate (PPTS, 26 mg) and *N*,*N*-dimethylformamide (DMF, 1 mL). The resulting solution was stirred at room temperature (rt) for 12 h, and then 5 mL of H_2_O was added. The reaction solution was extracted with 15mL of CH_2_Cl_2_, and the organic phase was concentrated under reduced pressure. The residue was purified by semipreparative HPLC (95% MeOH-H_2_O) to yield the acetonide **5e** (3.4 mg, t_R_ 5.7 min). Its structure was identified by ESIMS (Appendix A) and NMR data (Table 3 and Table 4, Figure 4 and Appendix A).

### 3.6. The Induced ECD Spectra of Compounds ***2***, ***5c***, and ***5d*** for Absolute Configuration

According to a published procedure [16,17], analytical pure DMSO was dried with 4 Å molecular sieves and was used to prepare 0.6 mg/mL of Mo_2_(OAc)_4_ solution. To three pieces of this solution (each 1 mL, 1.40 μmol), compounds **2** (0.5 mg, 1.86 μmol), **5c** (0.8 mg, 1.90 μmol), and **5d** (0.8 mg, 1.90 μmol) were respectively added and the first ECD spectra of the mixtures were recorded immediately. Then, ECD spectra were continuously recorded every 10 min until stationary. The inherent ECD spectrum was subtracted. The observed signs of the diagnostic bands in the region of λ_max_ 300–400 nm in the induced ECD spectra were correlated to the absolute configuration of the *ortho*-diol moiety.

(2*R*,3*R*,5*S*,9*R*,10*S*)-2,3,9,11-Tetrahydroxydrim-7-en-6-one (ustusol F, **1**): colorless oil; [α]^23^_D_ −56.0 (*c* 0.11, MeOH); UV (MeOH) λ_max_ (log *ε*) 232 (0.82) nm; ECD (1.76 mM, MeOH) λ_max_ (Δ*ε*) 336 (+8.4), 271 (−3.2), 240 (−41.3), 215 (−12.8) nm; IR (KBr) *ν*_max_ 3399, 2959, 1663, 1439, 1384, 1243, 1062, 1027 cm^−1^; ^1^H and ^13^C NMR see Table 1; HRESIMS *m*/*z* 285.1694 [M+H]^+^ (calcd for C_15_H_24_O_5_, 285.1697), or 283.1547 [M–H]^−^ (calcd for C_15_H_23_O_5_, 283.1551).

(2*R*,3*R*,5*R*,9*S*,10*R*)-2,3,11-Trihydroxydrim-7-en-6-one (9-deoxyustusol F, **2**): yellow oil; [α]^23^_D_ −56 (*c* 0.06, MeOH); UV (MeOH) λ_max_ (log *ε*) 238 (1.65) nm; ECD (1.87 mM, MeOH) λ_max_ (Δ*ε*) 334 (+6.8), 264 (−1.3), 240 (−18.3), 220 (−14.3) nm; IR (KBr) *ν*_max_ 3398, 2942, 1659, 1440, 1382, 1237, 1152, 1060, 983 cm^−1^; ^1^H and ^13^C NMR see Table 1; HRESIMS *m*/*z* 269.1751 [M+H]^+^ (calcd for C_15_H_24_O_4_, 269.1747).

(3*S*,5*R*,9*R*,10*R*)-3,11,12-Trihydroxydrim-7-en-6-one (ustusol G, **3**): colorless oil; [α]^23^_D_ −71 (*c* 0.04, MeOH); UV (MeOH) λ_max_ (log *ε*) 240 (1.60) nm; ECD (1.87 mM, MeOH) λ_max_ (Δ*ε*) 335 (+10.6), 265 (−2.6), 241 (−19.1), 205 (−71.7) nm; ^1^H and ^13^C NMR see Table 1; HRESIMS *m*/*z* 269.1750 [M+H]^+^ (calcd for C_15_H_24_O_4_, 269.1747).

(5*S*,6*R*,9*S*,10*S*,11*R*,2′*E*,4′*E*)-6-(11-Deoxy-11-hydroxystrobilactone A-6-yl)-5-carboxypenta-2,4-dienoate (ustusolate H, **4**): colorless solid; [α]^25^_D_ −96 (*c* 0.2, MeOH); UV (MeOH) λ_max_ (log *ε*) 264 (1.54) nm; ECD (0.64 mM, MeOH) λ_max_ (Δ*ε*) 264 (−6.2), 232 (−3.3), 205 (−11.1) nm; IR (KBr) *ν*_max_ 3434, 2953, 2926, 2856, 1684, 1640, 1460, 1398, 1310, 1260, 1208, 1136, 1028, 913 cm^−1^; ^1^H and ^13^C NMR see Table 2; HRESIMS *m*/*z* 391.1762 [M–H]^–^ (calcd for C_21_H_27_O_7_, 391.1762).

((5*S*,6*R*,9*S*,10*S*)-Strobilactone A-6-yl) (2*E*,4*E*)-6,7-dihydroxyocta-2,4-dienoate (ustusolate I, **5**): light yellow oil; UV (MeOH) λ_max_ (log *ε*) 265 (4.15) nm. ^1^H NMR (DMSO-*d*_6_, 500 MHz) *δ*_H_ 1.83 (d, *J* = 13.6 Hz, 1H, H-1*α*), 1.95 (dd, *J* = 4.4, 13.6 Hz, 1H, H-1*β*); 1.59 (m, 1H, H-2*α*), 1.47 (m, 1H, H-2*β*); 1.20 (td, *J* = 3.2, 13.1 Hz, 1H, H-3*α*), 1.34 (d, *J* = 12.3 Hz, 1H, H-3*β*); 2.00 (d, *J* = 5.0 Hz, 1H, H-5); 5.59 (brs, 1H, H-6); 5.79 (brs, 1H, H-7); 4.88 (dt, *J* = 2.3, 12.6 Hz, 1H, H-12*α*), 4.78 (d, *J* = 12.6 Hz, 1H, H-12*β*); 1.06 (s, 3H, H-13); 0.92 (s, 3H, H-14); 1.07 (s, 3H, H-15); 5.94 (d, *J* = 15.3 Hz, 1H, H-2′); 7.20/7.23 (m, 1H, H-3′); 6.40/6.44 (m, 1H, H-4′); 6.30/6.34 (m, 1H, H-5′); 3.85/3.97 (m, 1H, H-6′); 3.49/3.56 (m, 1H, H-7′); 0.94/1.02 (d, *J* = 6.2 Hz, 3H, H-8′); 5.02 (brs, 1H, HO-6′); 4.61/4.66 (brs, 1H, HO-7′); ^13^C NMR (DMSO-*d*_6_,125 MHz) *δ*_C_ 29.6 (CH_2_, C-1), 17.5 (CH_2_, C-2), 44.5 (CH_2_, C-3), 33.4 (C, C-4), 44.2 (CH, C-5), 65.8 (CH, C-6), 121.4 (CH, C-7), 136.6 (C, C-8), 73.2 (C, C-9), 37.3 (C, C-10), 174.4 (C, C-11), 68.3 (CH_2_, C-12), 18.3 (CH_3_, C-13), 32.2 (CH_3_, C-14), 24.4 (CH_3_, C-15), 165.51/165.50/165.49/165.47 (C, C-1′), 120.03/120.99/119.95/119.90 (CH, C-2′), 145.41/145.37/145.34/145.26 (CH, C-3′), 127.54/127.35/127.16/126.98 (CH, C-4′), 146.18/146.12/145.48/145.45 (CH, C-5′), 75.16/75.00/74.64/74.46 (CH, C-6′), 69.64/69.62/69.33/69.32 (CH, C-7′), and 19.34/19.26/18.26/18.24 (CH_3_, C-8′); ESIMS peak at *m*/*z* 419.1 for [M–H]^–^ and *m*/*z* 464.9 for [M + HCO_2_]^–^ (C_23_H_32_O_7_).

(2′*E*,4′*E*;6′,7′-*e**rythro*)-Ustusolate I (**5a**): light yellow oil; [α]^23^_D_ −35 (*c* 0.30, MeOH); UV (MeOH) λ_max_ (log *ε*) 268 (4.15) nm; ECD (0.60 mM, MeOH) λ_max_ (Δ*ε*) 255 (−8.3), 236 (−8.9), 208 (−21.2) nm; ^1^H and ^13^C NMR see Table 3 and Table 4; ESIMS *m*/*z* 421.2 [M+H]^+^ (C_23_H_32_O_7_).

(2′*E*,4′*E*;*ent*-6′,7′-*e**rythro*)-Ustusolate I (**5b**): light yellow oil; [α]^23^_D_ −42 (*c* 0.30, MeOH); UV (MeOH) λ_max_ (log *ε*) 261 (4.39) nm; ECD (0.60 mM, MeOH) λ_max_ (Δ*ε*) 256 (−11.0), 234 (−8.9), 209 (−21.4) nm; ^1^H and ^13^C NMR see Table 3 and Table 4; ESIMS *m*/*z* 421.2 [M+H]^+^ (C_23_H_32_O_7_).

(2′*E*,4′*E*,6′*R*,7′*R*)-Ustusolate I (**5c**): light yellow oil; [α]^23^_D_ −105 (*c* 0.30, MeOH; UV (MeOH) λ_max_ (log *ε*) 261 (4.42) nm; ECD (0.60 mM, MeOH) λ_max_ (Δ*ε*) 256 (−8.6), 234 (−8.5), 208 (−22.9) nm; ^1^H and ^13^C NMR see Table 3 and Table 4; ESIMS *m*/*z* 421.2 [M+H]^+^ (C_23_H_32_O_7_).

(2′*E*,4′*E*,6′*S*,7′*S*)-Ustusolate I (**5d**): light yellow oil; [α]^23^_D_ −79 (*c* 0.29, MeO; UV (MeOH) λ_max_ (log *ε*) 262 (4.36) nm; ECD (0.60 mM, MeOH) λ_max_ (Δ*ε*) 259 (−10.4), 234 (−8.2), 208 (−18.9) nm; ^1^H and ^13^C NMR see Table 3 and Table 4; ESIMS *m*/*z* 421.2 [M+H]^+^ (C_23_H_32_O_7_).

(2′*E*,4′*E*;6′,7′-*e**rythro*)-Ustusolate I-6′,7′-acetonide (**5e**): light yellow oil; [α]^22^_D_ −102 (*c* 0.16, MeOH); UV (MeOH) λ_max_ (log *ε*) 268 (4.15) nm; ECD (0.60 mM, MeOH) λ_max_ (Δ*ε*) 256 (−15.1), 232 (−13.8), 210 (−39.4) nm; ^1^H and ^13^C NMR see Table 3 and Table 4; ESIMS *m*/*z* 421.2 [M+H]^+^ (C_26_H_36_O_7_).

(5*S*,6*R*,9*S*,10*S*,2′*E*,4′*E*)-(Strobilactone A-6-yl)-5-carboxypenta-2,4-dienoate (ustusolate J, **6**): colorless solid; [α]^20^_D_ −280 (*c* 0.65, MeOH); UV (MeOH) λ_max_ (log *ε*) 265 (1.84) nm; ECD (0.64 mM, MeOH) λ_max_ (Δ*ε*) 261 (−11.4), 232 (−8.9), 207 (−23.2) nm; IR (KBr) *ν*_max_ 3400, 3320, 2950, 2928, 1658, 1615, 1460, 1385, 1290,1208, 1155, 1078, 970 cm^−1^; ^1^H and ^13^C NMR see Table 2; ESIMS *m*/*z* 459.5 [M–H]^–^ (C_21_H_26_O_7_).

(2*S*,5*S*,9*R*,10*S*)-2,9,11-Trihydroxydrim-7-en-6-one (ustusol B, **7**): light yellow solid; [α]^23^_D_ −140 (*c* 0.1, MeOH); UV (MeOH) λ_max_ (log *ε*) 252 (1.33) nm; ECD (1.87 mM, MeOH) λ_max_ (Δ*ε*) 335 (+11.9), 268 (−3.3), 241 (−66.9), 214 (−18.7) nm; IR (KBr) *ν*_max_ 3400, 3320, 2950, 2928, 1658, 1615, 1460, 1385, 1290, 1208, 1155, 1078, 970 cm^−1^; ^1^H and ^13^C NMR see Table 1; ESIMS *m*/*z* 269.2 [M+H]^+^, 291.2 [M+Na]^+^ (C_15_H_24_O_4_).

## 4. Conclusions

In summary, we identified four unpublished drimane sesquiterpenoids (**1**–**4**) and three published analogues (**5**–**7**) from the mangrove-derived fungus *Aspergillus ustus* 094102. Their structures including absolute configurations of **1**–**7** were determined by spectroscopic analysis, chemical reaction, and ECD spectra. Compound **5,** containing four stereoisomers of the chiral *ortho*-diol in the side chain, was further purified as the pure isomers **5a**–**5d** for the first time, among which the absolute configuration of the *threo*-6,7-diol (**5c** and **5d**) in the side chain was also determined by a dimolybdenum ECD method for the first time. In addition, the absolute configurations of the published compounds **6** and **7** were also resolved in this paper. Unresolved compound **5** displayed selective antiproliferation against CAL-62 and MG-63 tumor cells with the IC_50_ values of 16.3 and 10.1 µM, respectively, while the purified compounds **5a**–**5d** didn’t show activity.

## Figures and Tables

**Figure 1 marinedrugs-20-00408-f001:**
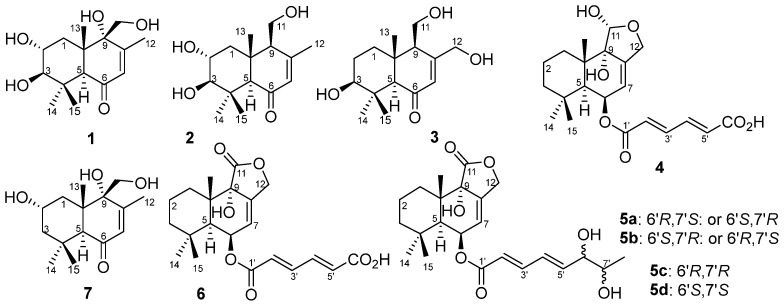
Structures of compounds **1**–**7** from *Aspergillus ustus* 094102.

**Figure 2 marinedrugs-20-00408-f002:**
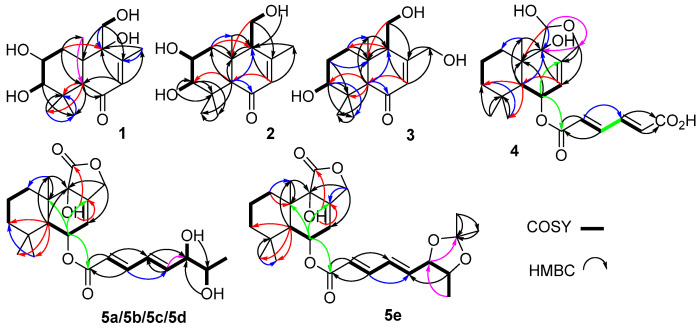
Key COSY and HMBC correlations of compounds **1**–**4** and **5a**–**5e**.

**Figure 3 marinedrugs-20-00408-f003:**
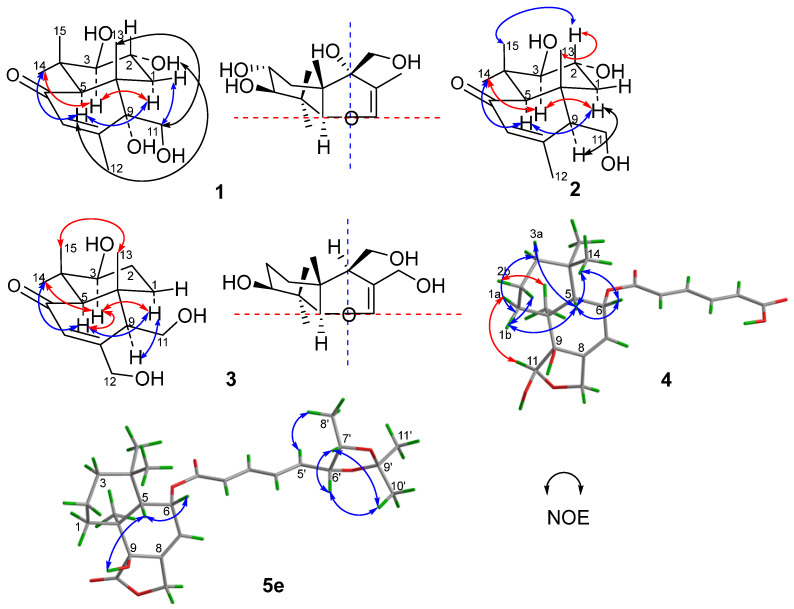
NOESY correlations of compounds **1**–**4** & **5e** and the octant rule for **1** and **3**.

**Figure 4 marinedrugs-20-00408-f004:**
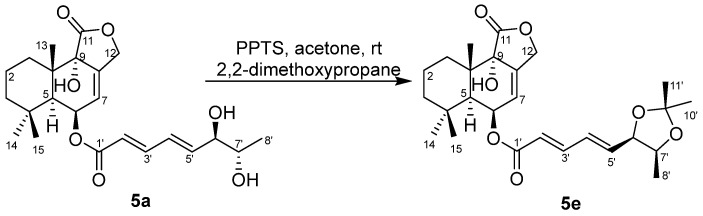
The preparation of acetonide **5e** from **5a**.

**Figure 5 marinedrugs-20-00408-f005:**
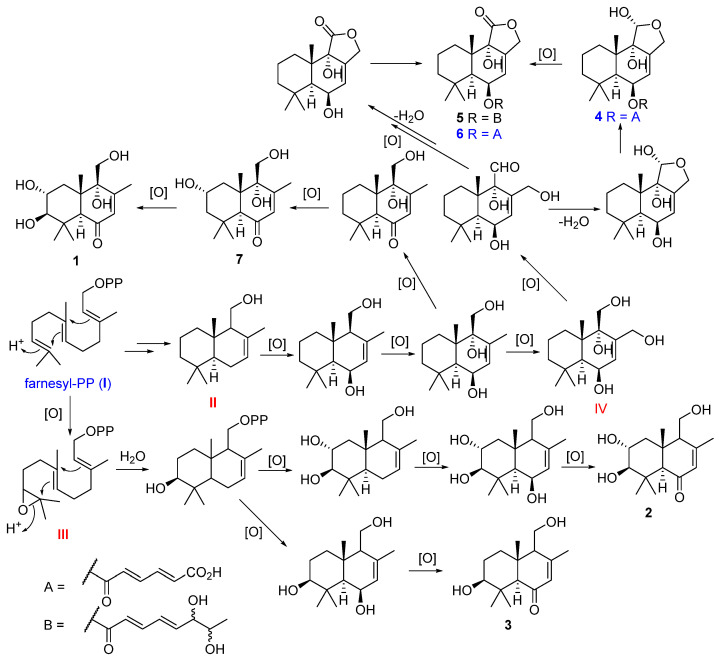
Proposed biosynthetic pathway for drimane sesquiterpenoids from *A*. *ustus* 094102.

**Table 1 marinedrugs-20-00408-t001:** ^1^H (500 MHz) and ^13^C (125 MHz) NMR Data for Compounds **1**–**3** and **7** (DMSO-*d*_6_, TMS, *δ* ppm).

Position	1		2		3		7	
	*δ*_C_, type	*δ*_H_, Mult. (*J* in Hz)	*δ*_C_, Type	*δ*_H_, mult. (*J* in Hz)	*δ*_C_, type	*δ*_H_, Mult. (*J* in Hz)	*δ*_C_, Type	*δ*_H_, Mult. (*J* in Hz)
1	38.6, CH_2_	*β* 1.69, dd (12.6, 4.6) *α* 1.85, dd (12.6, 12.1)	45.3, CH_2_	2.09, dd (12.6, 4.3) 1.33, dd (12.6, 12.1)	36.6, CH_2_	1.42–1.46, m 1.86–1.90, m	41.0, CH_2_	1.71–1.65, m 1.76–1.71, m
2	66.4, CH	3.46–3.48, m	66.1, CH	3.42–3.47, m	26.7, CH_2_	1.46–1.51, m, 2H	62.4, CH	3.68–3.72, m
3	81.6, CH	2.67, d (9.5)	81.6, CH	2.75, d (9.6)	76.8, CH	3.02, t (7.0)	51.7, CH_2_	0.96, t (11.9) 1.50, dd (11.9, 3.8)
4	37.8, C		38.0, C		37.5, C		33.4, C	
5	55.0, CH	2.81, s	61.6, CH	2.22, s	62.0, CH	2.15, s	54.7, CH	2.70, s
6	199.2, C		198.6, C		199.4, C		199.6, C	
7	128.2, CH	5.61, d (1.4)	127.9, CH	5.71, s	123.7, CH	5.96, s	128.1, CH	5.61, s
8	157.5, C		159.0, C		162.3, C		157.6, C	
9	74.6, C		57.1, CH	2.29, br s	55.1, CH	2.31, br s	74.6, C	
10	45.4, C		42.3, C		41.7, C		46.2, C	
11	61.9, CH_2_	3.53, d (11.5) 3.64, d (11.5)	57.9, CH_2_	3.61, dd (11.0, 5.0) 3.74, br d (11.4)	57.7, CH_2_	3.51–3.54, m 3.68, br d (10.9)	61.9, CH_2_	3.53, d (11.5) 3.64, d (11.5)
12	19.2, CH_3_	1.96, d (1.4)	21.5, CH_3_	1.98, s	61.3, CH_2_	4.19, d (18.1) 4.26, d (18.1)	19.2, CH_3_	1.98, s
13	16.7, CH_3_	1.04, s	16.7, CH_3_	0.88, s	15.8, CH_3_	0.80, s	18.9, CH_3_	1.08, s
14	29.3, CH_3_	1.14, s	28.7, CH_3_	1.12, s	28.5, CH_3_	1.10, s	33.8, CH_3_	1.14, s
15	19.1, CH_3_	1.04, s	16.5, CH_3_	1.03, s	15.5, CH_3_	0.99, s	22.7, CH_3_	1.03, s
2-OH		4.47, s		4.47, s				4.39, s
3-OH				4.50, s				
9-OH		5.06, s						5.02, s
11-OH		4.91, s		4.68, s				

**Table 2 marinedrugs-20-00408-t002:** ^1^H (500 MHz) and ^13^C (125 MHz) NMR Data for Compounds **4** and **6** (DMSO-*d*_6_, TMS, *δ* ppm).

Position	4		6	
	*δ*_C_, Type	*δ*_H_, Mult. (*J* in Hz)	*δ*_C_, Type	*δ*_H_, Mult. (*J* in Hz)
1	31.6, CH_2_	1.20–1.23, m 1.86, td (13.5, 4.1)	29.8, CH_2_	1.82, br d (13.5) 1.95, td (13.5, 4.1)
2	17.8, CH_2_	1.39–1.45, m 1.52–1.63, m	17.6, CH_2_	1.43–1.50, m 1.54–1.64, m
3	44.5, CH_2_	1.17–1.20, m 1.29–1.35, m	44.4, CH_2_	1.19, d (12.5) 1.34, br d (12.5)
4	33.3, C		33.5, C	
5	45.1, CH	2.07, d (4.6)	44.6, CH	2.01, d (4.7)
6	67.3, CH	5.58, br s	68.4, CH	5.79, br s
7	117.0, CH	5.49, d (2.3)	121.3, CH	5.60, br s
8	143.2, C		142.3, C	
9	76.4, C		73.3, C	
10	38.0, C		37.4, C	
11	97.4, CH	5.20, s	174.6, C	
12	65.8, CH_2_	4.08, d (13.0) 4.38, d (13.0)	66.6, CH_2_	4.78, d (12.7) 4.87, d (12.7)
13	18.6, CH_3_	1.12, s	18.5, CH_3_	1.05, s
14	32.7, CH_3_	0.91, s	24.5, CH_3_	1.05, s
15	24.5, CH_3_	1.06, s	32.3, CH_3_	0.91, s
1′	165.0, C		165.0, C	
2′	128.2, CH	6.39, dd (11.6, 2.9)	127.9, CH	6.33–6.43, overlap ^a^
3′	140.6, CH	7.32, dd (11.2, 2.9)	137.0, CH	7.27–7.35, overlap ^b^
4′	141.9, CH	7.29, dd (11.2, 2.9)	140.6, CH	7.27–7.35, overlap ^b^
5′	130.2, CH	6.35, dd (11.6, 2.9)	130.4, CH	6.33–6.43, overlap ^a^
6′	166.9, C		166.9, C	

^a^ Overlapping signals of H-2′ with H-5′; ^b^ Overlapping signals of H-3′ with H-4′.

**Table 3 marinedrugs-20-00408-t003:** ^1^H NMR Data for Compounds **5a**–**5e** (600 MHz, DMSO-*d*_6_, TMS, *δ* ppm).

Position	5a	5b	5c	5d	5e
	*δ*_H_, Mult. (*J* in Hz)	*δ*_H_, Mult. (*J* in Hz)	*δ*_H_, Mult. (*J* in Hz)	*δ*_H_, Mult. (*J* in Hz)	*δ*_H_, Mult. (*J* in Hz)
1a	1.83, d (13.6)	1.83, d (13.6)	1.84, d (13.6)	1.84, d (13.6)	1.83, d, (13.7)
1b	1.95, dd, (13.7, 4.3)	1.96, dd (13.7, 4.3)	1.96, dd (13.8, 4.2)	1.96, dd (13.7, 4.4)	1.96, dd (13.8, 4.4)
2a	1.48, dt (13.7, 3.9)	1.48, dt (13.7, 3.8)	1.48, dt (13.7, 3.8)	1.47, dt (13.7, 3.8)	1.45–1.49, m
2b	1.56–1.66, m	1.56–1.66, m	1.57–1.65, m	1.57–1.65, m	1.56–1.62, m
3a	1.21, td (13.3, 3.4)	1.21, td (13.3, 3.4)	1.20, td (13.3, 3.5)	1.21, td (13.3, 3.4)	1.18–1.23, m
3b	1.34, d (12.7)	1.34, d (12.7)	1.34, d (12.7)	1.34, d (12.7)	1.34, d (12.5)
5	2.02, d (4.9)	2.01, d (5.0)	2.01, d (5.0)	2.01, d (4.9)	2.01, d (5.0)
6	5.59, br s	5.59, br s	5.59, br s	5.59, br s	5.59, br s
7	5.79, br s	5.79, br s	5.79, br s	5.79, br s	5.79, br s
12a	4.79, d (12.6)	4.79, d (12.7)	4.79, d (12.6)	4.79, d (12.6)	4.78, d (12.7)
12b	4.88, dt (12.6, 2.4)	4.88, dt (12.6, 2.4)	4.88, dt (12.6, 2.4)	4.88, td (12.6, 2.4)	4.88, dt (12.6, 2.5)
13	1.06, s	1.06, s	1.06, s	1.06, s	1.06, s
14	0.92, s	0.92, s	0.92, s	0.92, s	0.92, s
15	1.07, s	1.07, s	1.07, s	1.07, s	1.07, s
2′	5.94, d (15.3)	5.94, d (15.3)	5.94, d (15.3)	5.94, d (15.3)	6.01, d (15.3)
3′	7.22, dd (15.3, 10.7)	7.22, dd (15.4, 10.7)	7.23, dd (15.3, 11.1)	7.23, dd (15.3, 11.1)	7.27, dd (15.3, 11.0)
4′	6.43, dd (15.3, 10.7)	6.42, dd (15.3, 10.8)	6.46, dd (15.3, 11.1)	6.45, dd (15.3, 11.2)	6.47, dd (15.2, 11.1)
5′	6.36, dd (15.3, 4.9)	6.34, dd (15.3, 5.1)	6.32, dd (15.3, 4.9)	6.30, dd (15.3, 5.1)	6.23, dd (15.2, 6.6)
6′	3.86, dd (10.2, 5.0)	3.84, dd (10.2, 5.1)	3.98, dd (10.2, 5.0)	3.96, dd (10.2, 5.1)	4.62, dd (12.8, 6.5)
7′	3.48, dq (11.6, 6.3)	3.48, dq (11.6, 6.3)	3.57, dq (11.6, 6.3)	3.57, dq (11.6, 6.3)	4.34, dq (12.8, 6.4)
8′	1.03, d (6.3)	1.03, d (6.3)	0.95, d (6.3)	0.95, d (6.3)	1.01, d (6.4)
9-OH	6.29, s	6.28, s	6.29, s	6.29, s	6.30, s
6′-OH	4.99, d (5.3)	5.00, d (5.2)	5.01, d (4.7)	5.02, d (4.7)	
7′-OH	4.60, d (5.3)	4.60, d (5.3)	4.66, d (4.7)	4.65, d (4.7)	
10′					1.28, s
11′					1.40, s

**Table 4 marinedrugs-20-00408-t004:** ^13^C NMR Data for Compounds **5a**–**5e** (150 MHz, DMSO-*d*_6_, TMS, δ ppm).

Position	5a	5b	5c	5d	5e
	*δ*_C_, Type	*δ*_C_, Type	*δ*_C_, Type	*δ*_C_, Type	*δ*_C_, Type
1	29.6, CH_2_	29.6, CH_2_	29.6, CH_2_	29.6, CH_2_	29.6, CH_2_
2	17.5, CH_2_	17.5, CH_2_	17.5, CH_2_	17.5, CH_2_	17.5, CH_2_
3	44.5, CH_2_	44.5, CH_2_	44.5, CH_2_	44.5, CH_2_	44.5, CH_2_
4	33.3, C	33.3, C	33.4, C	33.4, C	33.4, C
5	44.2, CH	44.2, CH	44.2, CH	44.2, CH	44.2, CH
6	65.8, CH	65.8, CH	65.8, CH	65.8, CH	66.0, CH
7	121.4, CH	121.4, CH	121.4, CH	121.4, CH	121.3, CH
8	137.2, C	136.6, C	136.6, C	136.6, C	136.7, C
9	73.2, C	73.2, C	73.2, C	73.2, C	73.2, C
10	37.3, C	37.3, C	37.3, C	37.3, C	37.3, C
11	174.4, C	174.4, C	174.4, C	174.4, C	174.4, C
12	68.3, CH_2_	68.2, CH_2_	68.3, CH_2_	68.3, CH_2_	68.3, CH_2_
13	18.3, CH_3_	18.3, CH_3_	18.3, CH_3_	18.3, CH_3_	18.3, CH_3_
14	32.2, CH_3_	32.2, CH_3_	32.2, CH_3_	32.2, CH_3_	32.2, CH_3_
15	24.3, CH_3_	24.3, CH_3_	24.3, CH_3_	24.3, CH_3_	24.3, CH_3_
1′	165.5, C	165.4, C	165.5, C	165.5, C	165.4, C
2′	119.9, CH	120.0, CH	119.9, CH	120.0, CH	121.4, CH
3′	145.5, CH	145.4, CH	145.3, CH	145.3, CH	144.6, CH
4′	126.9, CH	127.1, CH	127.3, CH	127.5, CH	129.2, CH
5′	146.2, CH	146.1, CH	145.4, CH	145.4, CH	140.4, CH
6′	75.0, CH	75.1, CH	74.4, CH	74.6, CH	77.7, CH
7′	69.6, CH	69.6, CH	69.3, CH	69.3, CH	73.5, CH
8′	19.2, CH_3_	19.3, CH_3_	18.3, CH_3_	18.3, CH_3_	16.0, CH_3_
9′					107.6, CH_3_
10′					25.4, CH_3_
11′					28.0, CH_3_

## Data Availability

This research is not involving humans or animals.

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
