# Peer review of "Sesquiterpenoids from the Mangrove-Derived *Aspergillus ustus* 094102"

_marinedrugs, 2022, doi:10.3390/md20070408_

Round 1

Reviewer 1 Report

The manuscript entitled “Sesquiterpenoids from the Mangrove-derived Aspergillus ustus 09102” written by Gui and co-workers describes on isolation and structural determination of new sesquiterpenoids from Chinese mangrove derived Aspergillus ustus. Contents involved in this paper are important information for marine drug development and worth publishing in marine drugs. However, there are many points to clarify or revise before publication. Then, the reviewer recommends this manuscript to be published after major revision. Please revise or respond on the following points.

1.      Table 2 should be reconstructed. The reviewer suggests that data of compound 7 should be included in Table 1 because of their closely related structures. And the authors should clarify correct structure of compound 7, which is different from the original paper of JNP2009. Had stereochemistry of compound 7 on C2 been already revised? Assignment of C13 and C15 are exchanged from the literature. Do the authors wat to revise data or stereochemistry of compound 7 in this manuscript? This is crucial point because new sesquiterpenoids are closely related to compound 7.

2.      Why 13C-NMR chemical sifts of C9 in compound 2 and 3 is 57 ppm and 55 ppm, respectively? They seem to be higher than usual sp3 carbons. This looks characteristic of these type of sesquiterpenids. Readers want to understand why? Please explain that.

3.      NOESY correlations in Figure 3 are hard to see. It is not friendly for readers. The reviewer strongly recommends reproduction of Figures as the molecules are. For example, C14methyl would be equatorial in chair 6-membered ring. Distance between 14-methl and H-5 looks far to make a NOESY cross peak in present figure.

4.      Wasn’t NOESY cross peak between H-2/13-methyl of compound 1 like compound 2? The authors don’t describe configuration of C2 of compound 1. This molecule should be described more detail based on obtained data.

5.      Table 1, H-NMR data of compound 3; make it clear there are 2 protons in this area of chemical shifts, or doubly indicate as other methylene protons in C1,C11,C12.

6.      Table 2 lacks of footnotes. What do a, b, c mean?

7.      Page 5, Line174; delete compound #6.

8.      The reviewer has a question that why the authors provide compound 5 as a single molecule at first, then described as a mixture made by four isomeric components. Finally, they were isolated by HPLC indeed. Or how do the authors think reported compound 5 was 5a?

9.      How do the authors explain why a mixture compound 5 exhibited significant cytotoxicity against two kind of tumor cell lines whereas each of the pure component was inactive.

Author Response

Dear reviewer

Thank you for your helpful suggestions to our manuscript “Sesquiterpenoids from the Mangrove-derived Aspergillus ustus 094102”. According to the your comments, we have checked and revised the manuscript carefully. The changes in manuscript were marked in yellow and the detailed responses were listed in the coverletter.

Kind regards

Weiming Zhu

Reviewer 2 Report

This manuscript entitled Sesquiterpenoids from the Mangrove-derived Aspergillus ustus 094102 by Gui et al describes the isolation and characterisation of novel marine derived natural products from the fungi. The introduction seems to be too short and does not show the significance of the work while the work is pretty novel in the field. The rest of the manuscript is well written and clearly explains the spectroscopic results including appropriate discussion. The results and discussion part are presented in good detail - not too long or too short.

The following are some comments I would like to make and/or suggest;

- Name - The name of the novel compound 4 is not consistent throughout the manuscript. The author should decide whethher to use ustusol H or ustusolate H. I recommend to use 'ustusolenic acid' instead as this name would better reflect the functional groups that appeared in the molecule of 4, and the structure of 4 is not relevant to other ustusol series so the new name might be better, and the compound 6 and 7 could also be named for convenience when your group wants to refer to these compounds in the future. For the compound 5, ustusolate I might not be a good name as the suffic -ate normally refers to either a salt or an acid group in the ester molecule where in this compound you want to refer to the ester 5. 

- The introduction part needs a rewrite and a major expansion to reflect the significance of your work with more reference on ustusol series please.

-Line 154 -  to the corresponding 

- Line 172-175 is confusing. I think the author may copy and paste this section from another part of the manuscript without proper readthrough.

- Conclusion needs an expansion too.

- For full characterisation, DEPT spectra would be necessary for all the novel compounds where these spectra are missing in SI.

- MS of compound 1 (Fig S1) seems strange as the M+H+ peak is not the major peak. Could you run this again? or include the elemental analysis for this compound?

- The HPLC spectra for all compounds should be included in SI to show the integrity and purity of the compounds.

- Fig S36, it is not clear why H5 can observe NOE with H13. The spectrum expansion around this peaks might be necessarily included.

- Fig S42, there are two C15. The peak at 32.2 should be labelled as 14.

- Fig S44, H2b does not show HSQC with any C?

- The authors may want to consider having a proper manuscript proofread by a native speaker.

Author Response

(The authors gave the same response as above.)

Reviewer 3 Report

From the fermentation broth of the mangrove-derived Aspergillus ustus 094102, the authors of this manuscript isolated four new drimane sesquiterpenoids (1−4) and three known ones (5−7). The structures of these drimane sesquiterpenoids were determined by HR-MS, 1D and 2D NMR, ECD, and chemical transformation. Bioactivity test showed that the compound 5 which is a mixture of isomers has weak antiproliferation activity toward the human tumor cells CAL-62 and MG-63 while all the other isolated drimane sesquiterpenoids including the purified isomers of compound 5 have no antiproliferation activity. Despite the authors have done an excellent job for the isolation and structure elucidation of four new drimane sesquiterpenoids, the novelty of the manuscript as a whole is not very significant. The structures of the isolated compounds are very similar to the known compounds and the bioactivity of the new isolated compounds is not good. Besides, the authors did not explain why compound 5 has weak antiproliferation activity while the purified isomers have no bioactivity at all. What’s more, the biosynthesis or potential relationships of these compounds were also not discussed. 

Author Response

(The authors gave the same response as above.)

Round 2

Reviewer 2 Report

The authors have promptly addressed all the queries that were raised during the first round of revision and therefore this manuscript is ready to be accepted after a thorough proofreading.

Reviewer 3 Report

Thanks to the authors' efforts for the revision. The manuscript is good now.